# DAREL: Training and Fine-Tuning Acceleration of Real and Hypercomplex Models

**Alexander Demidovskij**
National Research University Higher School of Economics
Nizhny Novgorod, Russia
monadv@yandex.ru

**Aleksei Trutnev**
National Research University Higher School of Economics
Nizhny Novgorod, Russia

**Artyom Tugaryov**
National Research University Higher School of Economics
Nizhny Novgorod, Russia

**Igor Salnikov**
Independent researcher

**Stanislav Pavlov**
National Research University Higher School of Economics
Nizhny Novgorod, Russia

## Abstract

Neural network training requires a lot of resources, and there are situations where training time and memory usage are limited. It makes specialized algorithms for training neural networks within the constraints of resource limitations an important and significant challenge. *Data Reduction with Losses* is a novel training data reduction method that operates with training samples based on losses obtained from a currently trained model or a pre-trained one. The proposed method can be used to train Deep Neural Networks for both Computer Vision and Natural Language Processing tasks in real and hypercomplex domains. When this method is applied to Large Language Models fine-tuning, *Data Reduction with Losses* is recommended to be combined with existing methods for Parameter-Efficient fine-tuning, such as LoRA. The training acceleration for ResNet18 is 2.03x, for Hypercomplex ResNet18 is 2.09x, GPT-2 Medium fine-tuning with *DAREL* on top of LoRA allows to achieve 1.43x acceleration with corresponding increase of BLEU score by 1.81 p.p. compared to baseline fine-tuning with LoRA method.

## 1  Introduction

Deep Learning (DL) has recently become the primary choice for numerous application domains, such as: computer vision [Shen et al., 2017], signal processing [Tuncer et al., 2020], natural language processing [Vaswani et al., 2017], robotics [Andrychowicz et al., 2020], autonomous driving [Janai

37th Conference on Neural Information Processing Systems (NeurIPS 2023).

et al., 2020] and reinforcement learning [Silver et al., 2018]. One of the clear reasons for DL to become so widespread among industrial applications is the constant growth of the computational capacity of new hardware and the enormous increase of data for analysis. It is believed that by 2025, the amount will be increased to 175 zettabytes [Rydning et al., 2018]. Moreover, modern datasets for training deep neural networks (DNN) also grow from 14.2M images in the ImageNet challenge in 2015 [Russakovsky et al., 2015] to 3B images in JFT-3B in 2022 [Zhai et al., 2022] increases the time required to train DNN. Futhermore, every hour that the GPU operates incurs a substantial cost to the environment due to its required power production, which leads to an increase in carbon dioxide emissions into the atmosphere [Strubell et al., 2019]. To investigate modern methods of DNN computing, recent studies have shown that hypercomplex neural networks outperform real-valued ones in terms of accuracy results [Pavlov et al., 2023], however, they underperform in training time. Hence, creating a cost-effective learning technology is the key challenge to solving the problem of a sharp increase in training time and computational resource consumption [Mirzasoleiman et al., 2020].

This paper makes the following primary contributions:

1. A novel two-stage method is designed to accelerate the pre-training of CNN and fine-tuning of LLM by implementing the importance sampling method based on losses information. According to our research, previous attempts were limited to applying importance sampling strategies to CNN models training only. We demonstrate acceleration of ResNet18 pre-training to 2.03x, Hypercomplex ResNet18 (ResNet18-HC) pre-training to 2.09x and GPT-2 Medium fine-tuning to 1.43x.

2. The concept of training budget for Computer Vision (CV) pre-training is introduced as a function of maximum GPU memory utilization and maximum training time. When pre-trained on ResNet18 a 1.26x acceleration is achieved with a time limitation by 80% and memory limitation by 70%.

3. Improvements to the state-of-the-art method for LLM fine-tuning [Hu et al., 2021] (LoRA) is delivered. Reported results include increase of BLEU score for GPT-2 Medium by 1.81 p.p. with 1.25x acceleration for E2E-NLG dataset.

The structure of the paper is as follows: Section 2 introduces a thorough overview of existing training acceleration methods. Section 3 proposes a new method of cost-efficient training. Section 4 contains a detailed evaluation of the proposed method. Section 5 draws the conclusion and highlights key directions for further research.

## 2 Background of study: overview

### 2.1 Problem definition

To discuss various approaches to CNN training acceleration, we need to first introduce certain terms that will be used throughout the paper. Let $w \in \mathbb{R}^m$ represent parameters of a model that we denote as $f_w$. Those weights are optimized to fit $f_w$ on a dataset $D = \{(x_i, y_i), i = 1, ..., N\}$, where $x_i \in \mathbb{R}^d$ is a data sample and $y_i$ is a ground truth label. Model prediction is denoted as $f_w(x_i)$ and means the output of a model on a given data sample $x_i$. The objective of a DNN is to minimize the loss between predictions given by a model and ground truth values in a whole training dataset. The loss is denoted as $l(f_w(x_i), y_i)$.

From a practical perspective, the training pipeline is usually organized in $E$ epochs, which are defined by the author of the training pipeline, usually based on their empirical experience. At each epoch, data from dataset $D$ is usually split into mini-batches, each containing $b$ samples. Then, at each epoch, model parameters are updated $N_b = \left\lceil \frac{|D|}{b} \right\rceil$ times. To sum up, most of the ideas behind training acceleration involve reducing of total training time $T$ required to achieve model convergence or complete a given number of epochs.

### 2.2 Key research directions in the field

There are numerous methods that aim at reducing training time and increasing model convergence speed. To begin with, quantization is considered as a prominent direction for accelerating both

training and inference of CNN and LLM. Quantization might be applied to the whole model by reducing the precision of parameters [Zhou et al., 2016] or gradients needed for parameter updates [Wen et al., 2017]. For the LLM field, most recent results include but are not limited to: PEQA [Kim et al., 2023], Alphatuning [Kwon et al., 2022], QLoRA [Dettmers et al., 2023].

The second direction of training acceleration research is reduction of trainable parameters of a model. For CNN, the well known approach is pruning, which is the removal of model weights depending on their influence on the resulting model accuracy [Liang et al., 2021]. For LLM, Parameter-Efficient Fine-Tuning is an emerging field that considers different ideas of making trainable fractions of original model weights. For example, by re-parametrizing some of them as suggested in LoRA [Hu et al., 2021] and its modifications [Zhang et al., 2023], [Zi et al., 2023].

The third direction is applying ideas of distributed computing to DNN training, which can be pipeline-parallel, model-parallel, or data-parallel [Wang et al., 2021]. With regard to LLM training, distributed training is almost a must with additional parallelization strategies such as sequence parallelism [Li et al., 2021].

The fourth direction is training data reduction. This is possible because of the assumption that not all samples from a dataset are equally important; some of them can be excluded from training without any visible impact on accuracy, and others require multiple iterations to get properly recognized by a DNN [Birodkar et al., 2019]. Within the domain of CNN training, there are three main approaches: dataset condensation or distillation [Cazenavette et al., 2022], core-set selection [Shim et al., 2021] and importance sampling [Jiang et al., 2019]. The main difference is what constitutes a reduced version of the training dataset. Data distillation methods suggest generating samples as entities synthesized according to some statistics learned from the original dataset. Core-set selection methods allow automatic selection of a subset from the original dataset, so that the resulting trained model achieves the same accuracy as that of the baseline within a given threshold, for example, 5%. The third approach within the data selection direction is importance sampling. This is performed by frequently selecting only those samples that would bring larger updates to weights during training, hereby becoming more informative from the standpoint of CNN training and LLM fine-tuning. However, these methods cause an additional overhead to the training process by either requiring a full forward pass for all samples in a dataset [Johnson and Guestrin, 2018] or training a separate network in parallel [Katharopoulos and Fleuret, 2017].

The state-of-the-art in this field is achieved with *Intellectual Data Selection* (*IDS*) and *Adaptive Online Importance Sampling* (*ADONIS*) methods [Demidovskij et al., 2023]. The *IDS* methodology proposes a way of filtering the training datasets by selecting diverse samples from each class in a labeled dataset. In order to evaluate diversity between samples, the pre-trained feature extractor ResNet18 is employed, and the similarity of samples is measured as the Euclidean distance between points in latent space with class prototype. The most common sample of a class is measured via K-Means clustering as a universal unsupervised approach for the evaluation of data patterns and similarities. Due to the pre-trained feature extraction, this approach is used as a preprocessing stage before training. *ADONIS* is aimed to reduce the number of backward passes by choosing samples from the available training data and constructing new training batches containing only the chosen elements. Samples for backpropagation passes are selected in a probabilistic manner. Each sample has a corresponding loss value, and based on that, its selection probability is obtained and followed by the final selection. Regarding LLM fine-tuning, an additional family of methods emerge, such as deduplication of samples in a dataset [Lee et al., 2021], [Tirumala et al., 2023].

The fifth direction is to accelerate pre-training and fine-tuning using new strategies [Zhang et al., 2016], [Dogra and Redman, 2020]. For example, applying the Hebbian learning rule and its modifications to the task of CNN pre-training [Lagani et al., 2021]: introducing mixed strategies (SGD+Hebbian learning) with reported training speedup up to 1.5x [Krithivasan et al., 2022], and implementation optimizations [Talloen et al., 2021]. As for innovative strategies for LLM fine-tuning, there is a vast scope for research and many attempts are made in terms of quicker convergence via new optimizers, such as LION [Chen et al., 2023] or Sophia [Liu et al., 2023].

# 3 Proposed method: Data Reduction with Losses

## 3.1 Budget-aware training algorithm

As mentioned above, in order to address the problem of training CNN and fine-tuning LLM in environments with limited resources, *Data Reduction with Losses* (*DAREL*) method is proposed. This method has a built-in support of the training budget $\mathbb{B}$ which has two key characteristics: total training time $t_{max}$ and maximum available memory $m_{max}$. Moreover, without any limitations, the budget is defined with respect to the full CNN pre-training of and LLM fine-tuning. Additionally, the training budget is also defined in a relative manner with $t$ as a ratio of full training time $T$, so that $t_{max} = t \cdot T$, and also $m$ as a ratio of memory required for full training $M$, so that $m_{max} = m \cdot M$.

*DAREL* is a two-stage training acceleration method that is designed to be budget-aware. The two stages of *DAREL* are called *offline* and *online*. The hyperparameters of both stages are automatically adapted for the given budget. The proposed method is based on the idea that reducing the number of samples due to a certain rule decreases the number of training steps, thus reducing the overall training time for a CNN and fine-tuning time for LLM.

## 3.2 Mathematical model

Initially, the memory requirements ($m$) amd time requirements ($t$) of a given budget are considered by the proposed method. The memory required to train a model is spent on loading and initializing the model itself with the training optimizer, criterion, loading and preprocessing training samples, etc. If the budget assumes that $m_{max}$ is smaller than the model itself, then training this model within this budget is not feasible. Therefore, the budget is considered realistic and assumes that model can be loaded to device. One of way to fit into training budget is reduce compute precision, for example from full precision do half precision. However, the precision is highly dependent on the selected training accelerator. As *DAREL* is designed to be agnostic to the training hardware, the only thing that remains is to reduce the training batch to a maximum memory that is required for training a model, as a peak memory is achieved while processing forward and backward operations on each batch.

To obtain a batch size $b$ that satisfies a given training budget $\mathbb{B}$ the following heuristic (Algorithm 1) is proposed. Let $\Upsilon$ denote the collection of time parameters and $\Psi$ memory parameters. While the memory usage for training a specific batch fits into the memory budget, the time $t_b$ and memory $m_b$ required for training batch of size $b$ is determined and collected. After obtaining batch training statistics, the batch size is heuristically evaluated with the $linear regression$ model by applying it to a specific memory budget $m_{max}$ and then rounding it down to a power of 2. The batch training time $t_b$ is then obtained as a time to train the batch with size $b$. The total training time $T_{new}$ could be approximated by (1), where $N_b$ is the number of batches and $E$ is the number of training epochs, $t_b$ is a time required for training batch with size $b$.

$$T_{new} = t_b \cdot N_b \cdot E. \tag{1}$$

While it is evident that reducing the batch size aids in controlling the required memory, it is also clear that $T_{new} \geq T$, because the approximate time for processing a single batch increases. In order to reduce this increased training time to match the time budget $t$, the following approach is elaborated. *DAREL* suggests total training time reduction via training data reduction during both *offline* (Algorithm 2) and *online* (Algorithm 3) stages. The *DAREL (Offline)* is based on the idea of accelerating training time by reducing the number of samples that a model needs to go through during the training process. Offline data reduction starts before training and online happens just-in-time of the training process.

*DAREL (Offline)* happens only once before the actual DNN training of fine-tuning. By reducing the number of samples in the training dataset, the overall epoch time (and not the step time) is decreased. We propose a selection of important samples from each class in a labeled dataset for filtering the training datasets. The $Mode$ of selection of these diverse samples is either selected as EASY or HARD for and selection ratio $\alpha$ is also determined. *DAREL (Offline)* leverages a pre-trained model ($\mu$) and builds a sorted list of samples based on loss information that is provided by that model in ascending order if $Mode$ is EASY, descending otherwise. Then, $1 - \alpha$ of samples with the smallest losses are removed from the training dataset. The selected samples are then added to the final dataset

---
**Algorithm 1** Automatic batch size detection for *DAREL*
---
1: $\mathbb{B}(t, m)$: given training budget
2: $\Gamma \leftarrow \{\}$: collection of processed batches
3: $\Upsilon \leftarrow \{\}$: collected time for processing batches of different sizes
4: $\Psi \leftarrow \{\}$: collected memory for processing batches of different sizes
5: $b \leftarrow 1$: initial batch size
6: **while** True **do**
7:    $t_b, m_b = trainBatch(b)$: collect time and memory required to train on a single batch of a given size
8:    $\Gamma \leftarrow \Gamma \cup b$: add new batch size
9:    $\Psi \leftarrow \Psi \cup m_b$: add memory required for processing current batch size
10:    $\Upsilon \leftarrow \Upsilon \cup t_b$: add time required for processing current batch size
11:    **if** $m_b \approx m_{max}$ **then**
12:       **break**
13:    **end if**
      $b \leftarrow b \cdot 2$
14: **end while**
15: $h(x) \leftarrow LinearRegression(X = \Psi, y = \Gamma)$: build batch size approximating model based on memory available
16: $b \leftarrow \lceil h(m_{max}) \rceil$: obtain approximated batch size within given training budget $\mathbb{B}$
17: $t_b \leftarrow \Upsilon_{\log_2 b}$: obtain approximated time for processing a single batch of size $b$
---

as demonstrated in Algorithm 2. Number of samples in each class from the dataset $D$ is reduced by its losses, keeping only $\alpha$ samples for training. In addition, changing the training dataset using the parameter $Mode$ is permitted.

---
**Algorithm 2** *DAREL (Offline)*
---
1: $Mode$: mode of selection {EASY, HARD}
2: $\alpha$: selection ratio
3: $\mu$: pre-trained model for a classification task, for example ResNet18
4: $Q = \{\}$: final dataset
5: **for** each class $D^i$ in $D$ **do**
6:    $\Theta^i = \{\mathbf{inf}\}$: losses for each sample in class $D^i$
7:    **for** each sample $j$ in $D^i$ **do**
8:       $\theta^i_j \leftarrow \mu(j)$: calculate loss for each sample
9:    **end for**
10:    $O^i \leftarrow sort(\Theta, Mode)$: sort samples w.r.t by loss values in ascending order if mode is EASY, descending otherwise
11:    $Q \leftarrow Q \cup \{O^i_j : j = 1, 2, \ldots, \lceil \alpha |D^i| \rceil\}$: add selected samples to final dataset
12: **end for**
---

The *DAREL (Online)* follows [Demidovskij et al., 2023]. Key hyperparameters of the *DAREL (Online)* are: selection ratio $A$ which defines how many samples from each batch are selected for backward, and loss update schedule $\eta$ which defines how regular loss information is updated for each sample. *DAREL* assumes automatic detection of these parameters values within the given training budget $\mathbb{B}$. During the online stage, history of losses $H$ is collected as double-ended queue of size $\gamma$ as described in (Algorithm 3). The number of warmup epochs is denoted as $\omega$. A histogram $\Phi$ is built to represent the losses.

**Algorithm 3** *DAREL (Online)*

1: $H \leftarrow deque(\gamma)$: history of losses as double-ended queue of size $\gamma$
2: $batch_{train} \leftarrow \{\}$: batch for training
3: $\eta, \eta \in \mathbb{N}, \eta \geq 1$: number of epochs when loss information is considered actual
4: $losses \leftarrow \{\}$: loss history
5: $E$: number of epochs to train
6: $\omega$: number of epoch to start *DAREL (Online)*
7: $last\_update \leftarrow 0$: number of last epoch when loss history was updated
8: $k \leftarrow 1$: training step
9: **while** epoch $i < E$ **do**
10:     **for** batch $batch_{train}$ from $D$ **do**
11:         **if** $i \leq \omega$ **then**
12:             $f_{w^{(k)}} \leftarrow trainBatch(b_{train}, f_{w^{(k-1)}})$
13:             $k \leftarrow k + 1$
14:             $batch_{train} \leftarrow \{\}$
15:             **continue**
16:         **end if**
17:         **if** $i - last\_update < \eta$ **then**
18:             $losses \leftarrow f_{w^{(k)}}(j)$
19:         **end if**
20:         **for** example $e$ from $batch_{train}$, loss $l$ from $losses$ **do**
21:             $H \leftarrow H \cup l$
22:             $\Phi \leftarrow buildHistogram(H)$: losses histogram representing losses
23:             $p \leftarrow PMF(\Phi, l)$: give preference to the most frequent losses
24:             $is\_chosen \leftarrow choose(p)$
25:             **if** $is\_chosen = 1$ **then**
26:                 $batch_{train} \leftarrow batch_{train} \cup e$
27:             **end if**
28:             **if** $|batch_{train}| = b$ **then**
29:                 $f_{w^{(k)}} \leftarrow trainBatch(b_{train}, f_{w^{(k-1)}})$
30:                 $batch_{train} \leftarrow \{\}$
31:             **end if**
32:         **end for**
33:     **end for**
34: **end while**

To limit total training time, *DAREL* provides a rule for defining the selection ratios of its stages: $\alpha$ and $A$ for offline and online stages, respectively. Note that the requested training budget and full training time are connected with the following way:

$$T \cdot t \geq T_{new} \cdot \alpha \cdot A \tag{2}$$

The default selection parameters of *DAREL* are as follows: $\alpha$ is equal to 0.8 and $A$ is equal to 0.1, as these values are seen to provide notable acceleration and superior accuracy results. By using default $\alpha$ value, the value of $A$ as a parameter of a *DAREL (Online)* method is recalculated:

$$A = \begin{cases} \frac{T \cdot t}{T_{new} \cdot \alpha}, & \text{if } T \cdot t \geq T_{new} \cdot \alpha \cdot 0.1 \\ 0.1, & \text{otherwise} \end{cases} \tag{3}$$

However, even the smallest possible value of $A$ cannot be enough to deliver the requested training budget. Therefore, tuning $\alpha$ parameter automatically is suggested in this case:

$$\alpha = \begin{cases} 0.8, & \text{by default} \\ \frac{T \cdot t}{T_{new} \cdot A}, & \text{if } T \cdot t < T_{new} \cdot 0.8 \cdot A \end{cases} \tag{4}$$

Finally, to promote training acceleration by making loss history updates less regular, it is suggested to select the parameter $\eta$ depending on parameter $A$. Although the accuracy of checking the final model may be reduced, it provides an advantage in the learning speed. This $\eta$ is obtained with:

$$\eta = \begin{cases} 1, & \text{by default} \\ 2, & \text{if } 0.5 < A < 0.8 \\ 3, & \text{if } A \leq 0.5 \end{cases} \tag{5}$$

After setting $\eta$, the value of $A$ is clipped from 0.1 to 0.8. By comparing this method with the state-of-the-art *IDS* and *ADONIS* algorithms, *DAREL* provides the following differences. Initially, the idea of performing training dataset reduction during the *DAREL (Offline)* is proposed based on loss information for each sample. Focusing on the execution of this method in *IDS*, it is noticed that self-supervised clustering is employed to identify class prototypes. The samples are then filtered based on either cosine or Euclidean distance between their latent representation and the latent representation of the cluster centroid. Samples obtained as a result are not selected from the center group in this method. Instead, *DAREL (Offline)* selection is performed in a more uniform manner, which results in smaller accuracy drops as described in Section 4. Secondly, *DAREL* is the only algorithm over selected alternatives that considers the training budget based on its perspective from time and memory consumption. Other methods, such as *IDS*, *ADONIS* are designed to occupy all resources allocated for the training process. Thirdly, although the *DAREL (Online)* employs the importance sampling strategy from *ADONIS* method, *DAREL* introduces automatic detection of optimal configurations for selection ratios that allow training performance to be more predictable.

## 4  Evaluation

**Training data**  For experimental evaluation the following publicly available datasets are used. CIFAR-100 dataset [Krizhevsky et al.] contains 60000 32x32 color examples for 100 classes, it is split into 50000 images for training and 10000 for testing. The E2E-NLG dataset [Dušek et al., 2020] contains 42000 training, 4600 validation text samples. The WebNLG dataset [Shimorina and Gardent, 2018] contains 69400 training and 872 validation text samples.

**Training protocol**  ResNet18 definition is taken from *torchvision v0.15.2*. GPT-2 Small, GPT-2 Medium and GPT-2 Large models and aforementioned datasets are taken from *transformers v4.33.2*. For the LLM fine-tuning with *DAREL (Offline)*, pre-trained GPT-2 Medium is used. For the CV training with *DAREL (Offline)*, ResNet18 pre-trained on CIFAR-100 dataset is used. The training protocol for ResNet18 on CIFAR-100 is as follows: the selected loss function is Cross Entropy, the optimizer is Stochastic Gradient Descent with a learning rate equal to 0.1, momentum is 0.9, and weight decay is 5e-4. The learning rate is dynamically reduced by a $\gamma_{lr}$ equal to 0.2 during training at specified milestones: $60^{th}$, $120^{th}$, and $160^{th}$ epochs. The learning rate schedule is stepLR with a step of 30. The data is loaded with 32 workers, with batch size equal to 128, randomly shuffled at each epoch. Each sample is processed with the following pipeline: random crop to size equal to 32 with padding equal to 4, then random horizontal flip, random rotation by angle, and sample normalization. The training protocol for Hypercomplex ResNet18 on CIFAR-100 is as follows: the selected loss function is Cross Entropy, the optimizer is Stochastic Gradient Descent with a learning rate equal to 0.1, momentum is 0.9, and weight decay is 5e-4. The cosine annealing is used as the learning rate scheduler. The learning rate is dynamically reduced by a $\gamma_{lr}$ equal to 0.1 during training at specified milestones: $60^{th}$, $120^{th}$, and $160^{th}$ epochs. The data is loaded with 32 workers baseline epoch of 200 with batch sizes equal to 128, which is randomly shuffled at each epoch and each sample is processed with random crop to size equal to 224. The following preprocessing parameters: *mean=[0.5070751592371323, 0.48654887331495095, 0.4409178433670343], std=[0.2673342858792401, 0.2564384629170883, 0.27615047132568404]* were used for sample normalization. The number of baseline training epochs on CIFAR-100 is 200. Finally, certain hardware optimizations are enabled: pinning memory for data loaders (*pin_memory=True*), benchmarking mode (*cudnn.benchmark=True*) and mixed-precision (full and half-precision floating point format).

The training protocol for GPT-2 Small, GPT-2 Medium, GPT-2 Large on E2E-NLG and WebNLG is as follows: the selected loss function is Cross Entropy, the optimizer is AdamW with a learning rate equal to 2e-4, Adam $\beta_1$ is 0.9, Adam $\beta_2$ is 0.999, and weight decay is 0.01, warmup steps is 500, label smoothing is 0.1. The number of baseline training epochs is 5, length penalty is 0.9, num beams is 10. Half-precision floating point format with Apex O2 optimization is enabled. The LoRA parameters are the following: rank $r$ equals to 4, LoRA $\alpha$ is 32, dropout is 0.1.

For training evaluation experiments, we are using the following hardware: CPU has 3.00GHz with 32 cores (frequency not fixed), OS: Ubuntu 20.04 LTS, kernel version: 5.4.0-136-generic; 1 GPU with 16GB memory card is used for the CV training and 2 GPU with 16GB memory cards for the LLM fine-tuning; training framework: PyTorch 1.13.0, programming language: Python 3.9. The memory consumption for each epoch is measured on each batch via *pynvml* v11.5.0. The energy consumption

Table 1: Budget-aware ResNet18 and Hypercomplex ResNet18 training on CIFAR-100 with *DAREL*.

| Model | Method | Batch | $\alpha$ | $A$ | $\eta$ | Epoch | Boost, x | Acc. drop, p.p. | Mem. cut, x | $CO_2e$ cut, x |
|---|---|---|---|---|---|---|---|---|---|---|
| ResNet18 | $\mathbb{B}(t=1, m=1)$ | 128 | 0.8 | 0.28 | 3 | 200 | **1.46** | **3.01** | **1.66** | **3.12** |
| | $\mathbb{B}(t=0.80, m=0.80)$ | 64 | 0.8 | 0.43 | 3 | 121 | 1.25 | **4.57** | 1.93 | 2.05 |
| | $\mathbb{B}(t=0.80, m=0.70)$ | 64 | 0.8 | 0.5 | 2 | 121 | **1.26** | 4.97 | 1.93 | 1.46 |
| | $\mathbb{B}(t=0.80, m=0.55)$ | 32 | 0.8 | 0.29 | 3 | 88 | 1.25 | 15.04 | **2.03** | **2.71** |
| | $\mathbb{B}(t=0.70, m=0.80)$ | 64 | 0.8 | 0.43 | 3 | 109 | 1.43 | **10.58** | 1.93 | 2.3 |
| | $\mathbb{B}(t=0.70, m=0.70)$ | 64 | 0.8 | 0.5 | 3 | 81 | 1.43 | 11.3 | 1.93 | 1.37 |
| | $\mathbb{B}(t=0.70, m=0.55)$ | 32 | 0.8 | 0.1 | 3 | 61 | **1.44** | 21.53 | **2.03** | **5.46** |
| | $\mathbb{B}(t=0.50, m=0.80)$ | 64 | 0.8 | 0.11 | 3 | 78 | 1.99 | 11.43 | 1.93 | 3.0 |
| | $\mathbb{B}(t=0.50, m=0.70)$ | 64 | 0.8 | 0.11 | 3 | 75 | 2.0 | **11.4** | 1.95 | 3.2 |
| | $\mathbb{B}(t=0.50, m=0.55)$ | 32 | 0.6 | 0.1 | 3 | 49 | **2.03** | 32.18 | **2.03** | **7.83** |
| ResNet18-HC | $\mathbb{B}(t=1, m=1)$ | 128 | 0.8 | 0.5 | 1 | 200 | 2.09 | 3.03 | 1.00 | 1.99 |
| | $\mathbb{B}(t=0.80, m=0.80)$ | 64 | 0.8 | 0.5 | 1 | 200 | 1.67 | 2.91 | 1.48 | 1.73 |
| | $\mathbb{B}(t=0.70, m=0.80)$ | 64 | 0.8 | 0.5 | 2 | 195 | 1.60 | 2.65 | 1.54 | 1.72 |

Table 2: Fine-tuning experiments of GPT-2 Small, GPT-2 Medium, GPT-2 Large on E2E-NLG with *DAREL*.

| Models | Method | Boost | BLEU↑ | TER↓ | METEOR↑ | NIST↑ |
|---|---|---|---|---|---|---|
| GPT-2 Small | LoRA | - | 67.3 | 66.43 | **75.82** | **6.39** |
| | LoRA + *DAREL* ($\alpha = 0.9$) | 1.12 | 68.22 | 65.59 | 73.79 | 6.06 |
| | LoRA + *DAREL* ($\alpha = 0.8$) | 1.26 | 69.52 | **64.92** | 74.79 | 6.09 |
| | LoRA + *DAREL* ($\alpha = 0.7$) | **1.43** | **69.53** | 65.62 | 73.59 | 5.93 |
| GPT-2 Medium | LoRA | - | 65.9 | 69.36 | **79.48** | **6.97** |
| | LoRA + *DAREL* ($\alpha = 0.9$) | 1.11 | 67.65 | 68.07 | 79.37 | 6.96 |
| | LoRA + *DAREL* ($\alpha = 0.8$) | 1.25 | **67.71** | **67.54** | 78.46 | 6.93 |
| | LoRA + *DAREL* ($\alpha = 0.7$) | **1.44** | 66.03 | 68.24 | 77.91 | 6.81 |
| GPT-2 Large | LoRA | - | 69.93 | 67.45 | **81.73** | 7.32 |
| | LoRA + *DAREL* ($\alpha = 0.9$) | 1.11 | **70.02** | **67.38** | 81.68 | **7.33** |
| | LoRA + *DAREL* ($\alpha = 0.8$) | 1.24 | 68.36 | 68.02 | 81.02 | 7.2 |
| | LoRA + *DAREL* ($\alpha = 0.7$) | **1.43** | 68.07 | 68.48 | 81.01 | 7.15 |

is estimated via *codecarbon* library v2.2.5. Finally, the LLM distributed fine-tuning is evaluated with *DeepSpeed* v0.10.3.

**Carbon footprint**   The carbon footprint is calculated according to (6) [Strubell et al., 2019], where $PUE$ denotes power usage effectiveness coefficient, $C_e^*$ denotes the carbon intensity factor, $t$ denotes computing time in hours, $p_c, p_d, p_g$ — average power draw from CPU, DRAM, and GPU accordingly. The $g$ is the number of GPUs used to train.

$$CO_2e = C_e^* \cdot \frac{PUE \cdot t \cdot (p_c + p_d + g \cdot p_g)}{1000} \tag{6}$$

Table 3: Fine-tuning experiments of GPT-2 Small, GPT-2 Medium, GPT-2 Large on WebNLG with *DAREL*.

| Models | Method | Boost | BLEU↑ | TER↓ | METEOR↑ | NIST↑ |
|--------|--------|-------|-------|------|---------|-------|
| GPT-2 Small | LoRA | - | **36.05** | **68.39** | **54.53** | **4.4** |
| | LoRA + *DAREL* ($\alpha = 0.9$) | 1.11 | 33.5 | 70.59 | 51.59 | 3.94 |
| | LoRA + *DAREL* ($\alpha = 0.8$) | 1.25 | 27.2 | 75.69 | 42.3 | 2.43 |
| | LoRA + *DAREL* ($\alpha = 0.7$) | **1.42** | 27.11 | 75.71 | 38.01 | 2.09 |
| GPT-2 Medium | LoRA | - | 47.48 | **65.57** | **63.71** | 7.42 |
| | LoRA + *DAREL* ($\alpha = 0.9$) | 1.11 | **48.07** | 65.83 | 63.7 | **7.51** |
| | LoRA + *DAREL* ($\alpha = 0.8$) | 1.24 | 46.51 | 68.56 | 60.52 | 7.42 |
| | LoRA + *DAREL* ($\alpha = 0.7$) | **1.42** | 41.32 | 71.97 | 51.73 | 6.29 |
| GPT-2 Large | LoRA | - | **53.68** | **60.54** | **69.35** | **8.36** |
| | LoRA + *DAREL* ($\alpha = 0.9$) | 1.11 | 51.66 | 62.99 | 64.81 | 7.94 |
| | LoRA + *DAREL* ($\alpha = 0.8$) | 1.25 | 48.94 | 65.04 | 62.77 | 7.4 |
| | LoRA + *DAREL* ($\alpha = 0.7$) | **1.42** | 48.53 | 68.4 | 62.9 | 7.86 |

**Key results** A detailed performance analysis of budget-aware pre-training acceleration for ResNet18 and ResNet18-HC is demonstrated in Table 1. *Accuracy drop* is measured in absolute percentage points and is calculated for each method relative to the corresponding baseline: ResNet18 training takes 2458 seconds with 75.86% Top-1 accuracy, ResNet18 training with *IDS* and *ADONIS* takes 1650 seconds with 73.06% Top-1 accuracy. *Boost* is measured as the ratio of time required to train on a corresponding baseline to the time required to train a model with the use of an acceleration algorithm. *Memory cut* is measured as the ratio of maximum memory usage of baseline training to training with the acceleration algorithm. The $CO_2e$ *cut* corresponds to a ratio of $CO_2$ generated during baseline training to training with the acceleration algorithm. Columns *Batch*, $\alpha$ and *A* represent the automatically computed *DAREL* parameters for the budgeting mechanism and acceleration approach.

Hypercomplex ResNet18 budget training was also compared with the baseline: ResNet18-HC training takes 12682 seconds with 77.47% Top-1 accuracy. Hypercomplex ResNet18 pre-training was performed with the budget $\mathbb{B}(t = 0.80, m = 0.80)$, which reduces memory consumption by 1.48x, experiences a boost of 1.67x. The least accuracy drop was achieved for $\mathbb{B}(t = 0.70, m = 0.80)$ which is noted to be 2.65p.p. We highlight $\mathbb{B}(t = 0.80, m = 0.80)$ due to significant boost, memory, $CO_2e$ reduction and least accuracy drop. As a result, the least GPU resource-intensive training process was performed with the budget parameter $m = 0.55$, which reduces memory consumption up to 2.03x for training. A faster training with a boost of 2.03x was performed with the budget $\mathbb{B}(t = 0.50, m = 0.55)$.

The *IDS* and *ADONIS* was evaluated with default parameters $\gamma = 120, \alpha_{IDS} = 0.8, \phi = resnet18, A_{ADONIS} = 0.3, l = 512, k = square, \sigma = 1, \eta = 1, \theta = 0.12$. The default budgeting parameters of *DAREL* are $t = 1, m = 1$. *DAREL* outperforms *IDS* and *ADONIS* in terms of 1.01x acceleration ratio, decrease of 1.69x in memory utilization and decrease of 1.58x in carbon dioxide emissions in budget $\mathbb{B}(t = 0.80, m = 0.80)$. This is possible due to automatically tuned training parameters of *DAREL* algorithm.

As for the LLM fine-tuning results obtained are for GPT-2 Small, GPT-2 Medium and GPT-2 Large on E2E-NLG and WebNLG datasets. These results are demonstrated in Table 2 and Table 3. Baseline GPT-2 Large fine-tuning on E2E-NLG is performed with a help of LoRA method: it takes 4198 seconds. Tables 2 and 3 show the E2E-NLG and WebNLG fine-tuning evaluation with different *DAREL* parameters. As for the E2E-NLG, the best 1.43x acceleration was performed with *DAREL* ($\alpha = 0.7$). The GPT-2 Small achieved 2.23 p.p. higher BLEU and 1.43x acceleration with *DAREL* ($\alpha = 0.7$) it has GPT-2 Medium reached 1.81 p.p. higher BLEU, 1.82 better TER, and 1.25x acceleration with *DAREL* ($\alpha = 0.8$). The GPT-2 Large achieved better BLEU, TER, NIST by 0.09 p.p., 0.07 p.p. and 0.01 p.p. correspondingly with 1.11x acceleration with *DAREL* $\alpha = 0.9$. For the WebNLG fine-tuning, the best 1.42x acceleration was performed with *DAREL* $\alpha = 0.7$. The GPT-2 Medium produced 0.59 p.p. better BLEU score and 0.09 better NIST score and 1.11 acceleration with *DAREL* $\alpha = 0.9$.

## 5 Conclusion

The industry is predisposed for the growth of neural network parameters and the increase of datasets size. It necessarily leads to an increase in the required computational resources as well as the training time. This paper makes the following primary contributions: it proposes a novel two-stage method that is designed, but not limited to accelerating the training of CNN and fine-tuning of LLM, introduces a budgeting training for CV pre-training as a combination of maximum GPU memory utilization and maximum training time, and delivers improvements to state-of-the-art method for LLM fine-tuning (LoRA). Training acceleration for ResNet18 is up to 2.03x and for Hypercomplex ResNet18 is up to 2.09x, while for fine-tuning with *DAREL* and LoRA allows to achieve 1.43x acceleration for GPT-2 Medium fine-tuning with corresponding increase of BLEU by 1.81 p.p. compared to LoRA based baseline fine-tuning. Also, the potential reduction of ecological impact is measured, and *DAREL* allows reduce carbon dioxide emissions by up to 7.83x for ResNet18 and 1.99x for Hypercomplex ResNet18. We define the following development horizons from the perspective of *DAREL* improvement: evaluation on a larger number of models and datasets, investigation of applicability to the Natural Language Understanding task and combinations with other fine-tuning acceleration methods such as (IA)[3] [Liu et al., 2022] and Prompt-Tuning [Lester et al., 2021].

## Acknowledgments

Authors express they gratitude to reviewers for valuable comments and sincerely thank Samuel Ragland Francis Nadine Suzanne for helpful proofreading. Work is partially supported by RSF grant 23-71-30008.

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
