# OpenReview forum: "DAREL: Data Reduction with Losses for Training Acceleration of Real and Hypercomplex Neural Networks"
_NeurIPS.cc/2023/Workshop/WANT — WANT@NeurIPS 2023 Poster_

### Official Review · Reviewer_HfaD · 2023-10-23
**Review of "DAREL: Data Reduction with Losses for Training Acceleration of Real and Hypercomplex Neural Networks"**

**Confidence:** 4

**Review:**

The authors propose a method to reduce computational time and memory use when training deep neural network models.

The paper presents an interesting approach but lacks in clarity.

Major problems:
 - The proposed method is not properly explained. How does it actually work? I can't be expected to read some other paper to understand what your method actually does. Even so, the proposed method has particularities that are not explained either.
 - What is the difference between the online and offline methods? This needs to be explained.
 - What does it mean to "heuristically evaluate the batch size b with the LinearModel"? When and how is the linear model actually used? What kind of linear models is it? A linear regression model? Why are there two names for it (LinearModel and RegressionModel)? Are they different or the same? If the former, what are the differences and how are they used? If the latter, why two names?
 - Algorithm 1. What are the inputs and the outputs? What does line 1 do? Why not use a do-while loop instead of while True? What is roman B? T and M are not sets, so what is actually supposed to be accumulated there? How do you quantify the \approx in the stopping criterion? What does the RegressionModel function do? What is done on line 15? What is done on line 16?
 - What is the blackboard D on line 161? A typo?
 - Equation 1 is stated on its own without any context, and not being part of any sentence.
 - Line 170: What is the gradient-based sample selection, and how do you use it?
 - Algorithm 2: What are the inputs and the outputs? Has the mode been explained in the text? Line 7: How do you assign something to a cardinality? Either way, the \Xi is never used. What is actually done on line 9? This is never explained. How are the cluster centers actually used? This is never properly explained. What is the \Theta without a superscript? What is actually done on line 12?
 - Where is the recommendation for \alpha=0.8 given, and motivated?
 - Where does the default value of 0.1 for A come from?
 - Equation 5 is stated on its own without any context, and not being part of any sentence.
 - What learning rate schedule is used?
 - Explain where the mean and std came from, and do you really nead that many digits?
 - Equation 6 is stated on its own without any context, and not being part of any sentence.
 - Line 266: A result is that you propse particular settings? How do you motivate it?

Minor problems:
 - Most citations are written in text inline format (\citet), when they should be in parentheses (\citep).
 - What does [.] mean on line 64? Typically that's the round to nearest integer function, but you can't round the number of parameter updates up since there are no more samples to use.
 - Line 169: Use "---" instead of "-" there.
 - Language: Many typos, spelling, and grammar errors. Go through it thoroughly.

---

### Official Review · Reviewer_PnFt · 2023-10-24
**Problems with legibility**

**Confidence:** 2

**Review:**

My apologies to the authors but I do not know what the method is from reading the paper. Algorithm 1 appears to describe a method for approximating the batch size required before training and Algorithm 2 a method for ranking the datapoints to be processed, I assume for dataset selection. This seems to be presenting some complete training system with many moving parts that is not a very good fit for an academic paper, there is no clear hypothesis or finding to focus on or test by experiment.

The paper does not begin describing the method until page 4 and the description is very difficult to follow in the text. Algorithms 1 and 2 are not sufficient to explain what the purpose of the operations are even with the pseudocode.

Experiments demonstrate training speedups on CIFAR-100 of between 1.5 and 2. State of the art methods for training quickly on these datasets, for example the [DAWNBench](https://dawn.cs.stanford.edu/benchmark/index.html#cifar10-train-time) competition on CIFAR-10, can typically achieve much greater speedups than this, up to orders of magnitude. For example, [Super-Convergence](https://arxiv.org/abs/1708.07120) demonstrated training similar models 10x faster on CIFAR-100/Imagenet.

The GPT2 Finetuning results are more significant but I do not know how to interpret them because I don't understand the method and there is no comparison to competing methods.

Pros:

- May reduce the cost of training large models based on experimental results presented

Cons:

- Presentation:
    - The method itself is not discussed until page 4 and I was not able to figure out what it is from the description
    - Parentheses are not used for citations
    - Paragraphs are extremely long in some cases with no clear thread of argument
- Results do not compare to any similar method

---

### Official Review · Reviewer_TcHN · 2023-10-25
**An outstanding submission, great contribution to the workshop.**

**Confidence:** 4

**Review:**

Overview

The paper presents a two-stage method for reducing the number of samples in a dataset to accelerate the training of deep neural networks within conditions of an introduced concept of training budget (memory and time). Before training, the method selects the hardest and diverse samples across classes by utilizing a pre-trained model loss. During the training, the method utilizes the existing importance sampling method (ADONIS) with introduced automatic budget-aware detection of optimal parameters. The experiments provided include a classification task (CIFAR-100) and two natural language generation tasks (E2E-NLG, WebNLG) and the authors present a detailed experimentation report and results analysis.

Strengths (originality, quality, clarity, significance)
- The method proposed is a significant contribution to the topic of data selection for training acceleration, and while it utilizes some common ideas (loss-based filtering, importance sampling) the authors made considerable improvements by introducing the concept of budget and performing reduction methods both before and during training.
- The authors provided a great related work section, which presents 5 key research directions in training acceleration each of them touching different faces of the problem, including fundamental works and very recent ones.
- The paper provides a clear methodology and all details of experiments, including versions of libraries used, which makes the work reproducible (to some extent).

Weaknesses
- The tables of results miss significance testing, the quality difference of the proposed method to a baseline counts in very little numbers, e.g., GPT2-M tuning for WebNLG, 48.07 vs 47.48 BLEU score and others. For showing quality maintenance random seed runs might not be necessary, but for showing "improvement" in quality, as it is stated in the list of contributions, significance testing should be done, which will definitely improve the soundness of this result.

Conclusion: the submission is outstanding and brings clear contributions.

---

### Meta-Review · Area_Chair_uXmY · 2023-10-27

**Recommendation:** Accept (Poster)
**Confidence:** 2

**Metareview:**

The paper presents an algorithm for data reduction. While reviewer 1 and 3 find the paper and results interesting, it doesn't seem to be sufficiently well explained. Both reviewer 2 and 3 see major issues with the legibility of the article. As the idea seems to still be somewhat interesting and relevant to the workshop, I recommend the manuscript for a Poster.

---

### Decision · Program_Chairs · 2023-10-28

**Decision:**

Accept (Poster)

**Comment:**

We thank the authors for their time and contribution to WANT and we are pleased to share that after the reviewing process the paper has been accepted. Congratulations! We encourage the authors to consider reviewers' feedback for the improvement of the camera-ready version. We hope to see you in person at the workshop and brainstorm on efficient training research together!